# Carbon Nanomaterials: Emerging Roles in Immuno-Oncology

**DOI:** 10.3390/ijms24076600

**Published:** 2023-04-01

**Authors:** Bbumba Patrick, Tahira Akhtar, Rubina Kousar, Chih-Ching Huang, Xing-Guo Li

**Affiliations:** 1Graduate Institute of Biomedical Sciences, China Medical University, Taichung 406040, Taiwan; bumbax22@gmail.com (B.P.); tahiraramzan205@gmail.com (T.A.); rubimansib@gmail.com (R.K.); 2Research Center for Cancer Biology, China Medical University, Taichung 406040, Taiwan; 3Institute of Biochemistry and Molecular Biology, College of Life Sciences, China Medical University, Taichung 406040, Taiwan; 4Institute of Translational Medicine and New Drug Development, China Medical University, Taichung 406040, Taiwan; 5Department of Biological Science and Technology, China Medical University, Taichung 406040, Taiwan; 6Department of Bioscience and Biotechnology, National Taiwan Ocean University, Keelung 20224, Taiwan; huanging@ntou.edu.tw

**Keywords:** carbon nanomaterials, immunotherapy, cancer

## Abstract

Cancer immunotherapy has made breakthrough progress in cancer treatment. However, only a subset of patients benefits from immunotherapy. Given their unique structure, composition, and interactions with the immune system, carbon nanomaterials have recently attracted tremendous interest in their roles as modulators of antitumor immunity. Here, we focused on the latest advances in the immunological effects of carbon nanomaterials. We also reviewed the current preclinical applications of these materials in cancer therapy. Finally, we discussed the challenges to be overcome before the full potential of carbon nanomaterials can be utilized in cancer therapies to ultimately improve patient outcomes.

## 1. Introduction

Traditionally, the major theranostics for cancer patients have been surgery, chemotherapy, and radiation [1]. With a better knowledge of the relationship between oncology and immunology, it is now possible to use patients’ immune systems to fight cancer [2]. Systemic toxicity, cancer recurrence, and metastasis, on the other hand, have an impact on patients’ prognoses. Fortunately, current advances in immuno-oncology have recognized that prospective treatment strategies should address this unfulfilled need to prevent aggressive cancer relapses [3]. Cancer immunotherapies that can induce immunological memory have demonstrated a lasting inhibitory effect on cancers’ growth, recurrence, and metastasis [4]. Immune checkpoint blockade (ICB) and chimeric antigen receptor T (CAR-T) cell treatment are examples of cancer immunotherapies that have increased overall survival in a subgroup of patients, particularly in those with hematological tumors. However, only a subset of patients and/or certain cancer types respond favorably to immunotherapy, mainly owing to the immunosuppressive milieu of the solid tumor and immune resistance to mono-therapeutics [5]. Moreover, the systemic delivery of immunotherapeutic medicines might result in severe autoimmune toxicities. To boost the activity of the immune response, innovative drug delivery techniques with improved targeting and tumor microenvironment (TME)-modifying capabilities are critically needed for cancer immunotherapy. Nanomaterials have gained much attention as prospective cancer therapy options because they can integrate multifunctional components (such as immunostimulants and chemotherapeutic medicines) and exhibit distinctive physicochemical features [6,7,8]. All nanomaterials made of carbon atoms are referred to as carbon-based or carbon nanomaterials (CNMs), which have received a tremendous amount of attention in recent years. In this review, we provide a mechanism-based summary of CNMs in the antitumor immune response, and highlight the benefits and limitations of CNMs for improving the immunomodulatory effect of current cancer therapy.

## 2. CNMs: Classification and Structural Characteristics

Typically, based on their dimensional and geometrical structure, CNMs can be classified into four categories: 0D (zero-dimensional) CNMs (fullerenes, particulate diamonds, and carbon dots), 1D (one-dimensional) CNMs (carbon nanotubes, carbon nanofibers, and diamond nanorods), 2D (two-dimensional) CNMs (graphene, graphite sheets, and diamond nanoplatelets), and 3D (three-dimensional) CNMs (nanostructured diamond-like carbon films, nanocrystalline diamond films, and fullerite) [9]. Carbon nanostructures can be tube-shaped (single-walled nanotubes (SWCNTs) and multiwalled nanotubes (MWCNTs), horn-shaped (nanohorns), or spheres or ellipsoids (fullerenes). Fullerenes are carbon molecules or molecular forms of carbon, whereas graphene is a single sheet of carbon atoms [10]. Nevertheless, carbon nanomaterials have been successfully manipulated to generate nanoscale carbon particles (carbon dots) [11] and graphene-based materials known as graphene quantum dots (GQDs) capable of biological uses [12]. Furthermore, depending on their carbon hybridization, CNMs can exhibit a wide range of crystallinity, including various proportion of sp^2^ and sp^3^ carbon bonds. CNMs are very flexible due to their unique characteristics, which allow them to form alternative covalent or noncovalent bonds with other carbon atoms or elements to diversify their functionalization [13]. Table 1 summarizes the categorization and basic structural features of CNMs based on their dimensions. 

## 3. Functionalization of CNMs in Targeting Cancer

The functionalization of CNMs is a popular method for tuning the hydrophilicity of carbon nanostructures while also imparting biocompatible properties. This process involves grafting functional groups onto the surface of CNMs, resulting in the development of stable structures. Notably, biomedical applications such as immuno-oncology need total biocompatibility in order to avoid undesirable immune system responses [15]. The functionalization of CNMs has the potential to change their physical and chemical characteristics, as well as increase their therapeutic efficacy and bioactivity, reduce the immune response, and enable targeted drug delivery [16]. This implies that the ease of chemically modifying CNMs provide another layer of capacity to create new systems that can be adapted for specific interventions in immuno-oncology. CNTs, graphene, CDs, and fullerenes have been reported to potentially increase diagnostic accuracy for tumors and infectious diseases. Moreover, most nanoparticles can deliver medications to tumor cells either passively (through selectively enhanced permeability and retention of the tumor’s vasculature) or actively (by endocytic pathways). Molecules capped with different ligands bind to cell receptors and enter the cells via endocytosis, delivering a larger concentration of the drug to the interior of a cancer cell while causing less cytotoxicity to normal cells [17].

For instance, the anticancer drug doxorubicin (DOX) and the magnetic resonance imaging (MRI) contrast agent gadolinium-diethylenetriamine penta-acetic acid (Gd-DTPA) can be loaded into an asparagine-glycine-arginine (NGR) peptide-modified SWCNTs system to enter and accumulated within tumor cells, allowing chemotherapy and tumor diagnosis to be combined in one system [18]. Similarly, a photo-theranostic agent based on sinoporphyrin sodium (DVDMS)-loaded PEGylated graphene oxide (GO-PEG-DVDMS) was developed. This GO-PEG vehicle greatly boosted the efficiency of DVDMS in accumulating in tumors and the effectiveness of photodynamic therapy (PDT) in U87MG human glioma tumor cells in vivo [19]. Notably, Moon et al. demonstrated the in vivo destruction of solid malignant tumors using polyethylene glycol-coated single-walled carbon nanotubes (PEG-SWCNTs) coupled with near-infrared (NIR) irradiation. The photothermal impact of PEG-SWCNTs was investigated in nude mice with human epidermoid oral carcinoma KB tumor cells. Tumors were completely destroyed in the mice treated with PEG-SWCNTs followed by NIR irradiation [20]. Intriguingly, the benefits of functionalized CNMs may not only be beneficial for solid tumors but also hematological malignancies [21]. For example, polyethylene glycol-coated discrete MWCNTs (PEG-dMWCNTs) were designed with strong binding of DOX and targeting molecules (alendronate) in mice with Burkitt’s lymphoma, which decreased the cancer burden and enhanced survival. PEG-dMWCNTs therefore offered a potential novel nanocarrier platform for the safe delivery of drugs for hematological malignancies [21]. These examples demonstrated that CNMs are crucial in the field of cancer theranostics because they provide several benefits such as enhanced detection, tumor-specific drug delivery, and less fatal effects on normal tissues during cancer treatment [22]. Furthermore, CNMs are inexpensive, stable, and biodegradable, and have good photothermal conversion in the NIR range, making them promising candidates for photoacoustic imaging and photothermal therapy (PTT) [23]. The photothermal heat can stimulate dying tumor cells to release antigens, pro-inflammatory cytokines, and immunogenic intracellular substrates, promoting immune activation during immunogenic cell death (ICD). In one study, dendritic cells (DCs) collected the released damaged-associated molecular patterns (DAMPs) and tumor-associated antigens (TAAs), then processed and transmitted them to the adaptive immune cells to trigger antitumor immune responses [24]. Therefore, a deep understanding of the immunomodulatory activities of CNMs can help develop more effective therapeutics by harnessing the immunoregulatory effects of CNMs to achieve durable efficacy following cancer treatments.

## 4. CNMs versus Lipid-Based Nanomaterials: Focusing on Immuno-Oncology

The majority of cancer immunotherapies focus on the administration of tumor-associated antigens (TAAs) and tumor-specific antigens (TSAs). When the encoded antigen is translated to proteins in the cytoplasm of antigen-presenting cells (APCs), it can trigger an antigen-specific immune response. APCs process these proteins and display them on major histocompatibility complex (MHC) Class I (MHC I) to CD8+ T lymphocytes, promoting cell-mediated immune responses. Additionally, the MHC II trafficking signals produced from lysosomal proteins can also induce a supportive CD4+ T helper cell response if fused with an mRNA-encoded antigen, which is important in cancer immunotherapy [25]. This implies that combining nanoparticles with adjuvants may boost the activation of the immune response against cancer if effectively delivered to the target cells [26].

Nanomaterial-based delivery techniques have previously provided a suitable solution to cancer immunotherapy’s essential challenges [27]. The primary hurdles for cancer immunotherapies can be ascribed to the lack of delivery mechanisms that can keep therapeutic payloads accessible to their targets [28]. However, due to their extensive tunable functional groups and drug-carrying abilities, nanomaterials can enable tailored drug delivery to tumor locations or immunological organs. By reacting to internal or external stimuli, they can perform specific functionalities such as drug integration, effective biological barrier penetration, accurate administration of immunomodulators, and regulated release to enable effective tumor immunotherapy [27].

Nonetheless, despite biomedical nanotechnology’s substantial contribution to health care management, major efforts are being made to solve difficulties such as their poor repeatability, specificity, effectiveness, and cost. In addition, the drug nanocarriers used should be biocompatible and stimulus-responsive in order to execute regulated drug delivery and discharge, including in the brain [29]. As a result, various classes of nanomaterials have been created to address these inefficiencies. Lipid-based nanoparticles, CNMs, polymer-based nanomaterials, and metal-based nanomaterials are some examples of nanomaterial groupings. Because several lipid-based nanoparticles have previously been chosen for clinical studies, it is worthwhile to compare them with CNMs in terms of immuno-oncology.

Liposomes are examples of lipid-based nanomaterials. They are primarily made up of phospholipids that can create both unilamellar and multilamellar vesicles, which enable them to transport and distribute hydrophilic, hydrophobic, and lipophilic drugs, as well as entangle hydrophilic and lipophilic molecules in the same system [30]. In contrast, CNTs, which are one of the most common examples of CNMs, are highly insoluble and must often be chemically treated before they can be dispersed in various liquids. Their insolubility in the most common dispersing agents, such as surfactants or polymers, results in a colloidal dispersion rather than a solution, which may limit their use in drug delivery in immunotherapy [31]. 

Another hurdle posed by CNMs is the biodistribution and pharmacokinetics of nanoparticles, which are influenced by a variety of physicochemical properties, such as their shape, size, chemical composition, aggregation, solubility, and functionalization. Particles smaller than 100 nm have been reported to increase hazardous effects to the lung, the evasion of typical phagocytic defenses, structural changes in proteins, activation of inflammatory and immunological responses, and possible redistribution from their site of accumulation [32]. However, the key advantages of nonstructured lipid nanostructures which were developed from structured lipid nanostructures include the ability to be loaded with hydrophilic and hydrophobic drugs, to be surface-modified, to allow for site-specific targeting and controlled release of the drug, and their low in vivo toxicity. However, there are significant drawbacks as well, such as drug ejection following polymorphic transition of the lipid from the nanocarrier matrix during storage, and poor loading capacity [33].

Using resonance Raman spectroscopy, the oxidative stability of three typical 1D CNMs, including linear carbon chains, CNTs, and graphene nanoribbons, were systematically studied and found to be thermally stable up to 500 °C [34]. However, Holm et al. reported that lipid-based formulations were less optimized and could contain traces of peroxides with the potential to catalyze their degradation. This degradation could create difficulties in achieving the shelf-life of the formulation for supporting preclinical and clinical trial studies. Furthermore, there are no documented antioxidants that can be added to the formulations to prevent this degradation, which indicates that a relationship between stability and lipid-based nanomaterials needs to be clarified [35]. Notably, lipid-based nanoparticles possess advantages which include high temporal and thermal stability, high loading capacity, ease of preparation, low production costs, and large-scale industrial production, since they can be prepared from natural sources [36].

The industry’s expertise, in addition to its scientific and manufacturing platforms, could greatly influence the choice to utilize either lipid-based formulations or a CNM formulation. Unfortunately, this information has not been clearly spelt out. As a result, the public domain does not fully explain the determining relevant composition of the formulation of the molecule. A detailed investigation of successful events and the knowledge on how to establish their composition might help to lower perceptions of risk. The essential in vitro features in the design of these formulations for identifying the optimum composition of the formulation, as well as appropriate quality methodologies, must be examined for in vitro and in vivo applications. More research is needed to determine which animal species should be utilized to explore individual formulations [37]. 

Nonetheless, the unique characteristics of lipid-based nanomaterials and CNMs can enable them to selectively modulate critical signaling pathways inside diverse immune cell populations through their material compositions, shapes, or contact alterations to elicit significant antitumor effects [38]. For instance, carbon- and lipid-based nanoparticles may both be encapsulated with antigens and used for systemic delivery into APCs, similar to DCs. DCs then stimulate antitumor T cell responses by antigen translation and cross-presentation [39,40]. Nanoparticles imprinted with tumor antigens can increase the transport to APCs in lymphoid organs, leading to better DC maturation and T cell-mediated tumor death. Aside from delivery, nanoparticles can induce anticancer immune cell phenotypes [41]. Carboxylated MWCNTs (MWCNTs-COOH), for example, can limit tumors from spreading by modulating the polarization of macrophages [42].

## 5. Immunomodulation of CNMs in Preclinical Oncological Studies

Immune dysfunction is linked to an increased risk of certain cancers [43]. This indicates that appropriate immune activation may protect against some cancers. Furthermore, tumor cells are genetically unstable and may be difficult to target utilizing particular therapy regimens due to the tumor’s resistance [44]. This implies that the immunomodulatory activities of CNMs can be leveraged to minimize the progression of cancer [Figure 1]. Through a combinational application with chemotherapy, phototherapy or radiotherapy, CNMs may elicit the activation of T cells against tumor cells and enhance anticancer efficacy with lesser toxicity.

### 5.1. CNTs

Hassan et al. demonstrated that effective tumor elimination necessitates a stronger antitumor immune response. They used MWCNTs as tumor antigen nanocarriers to deliver immunoadjuvants such as cytosine-phosphate-guanine oligodeoxynucleotide (CpG) and anti-CD40 Ig (CD40) with the model antigen ovalbumin (OVA) to elicit an immune response against OVA-expressing tumor cells. The MWCNTs boosted the CpG-mediated adjuvanticity, as evidenced by the dramatically higher OVA-specific T cell responses in vitro and in C57BL/6 mice. MWCNTs significantly increased the efficacy of coloaded OVA, CpG, and CD40 to prevent the proliferation of OVA-expressing B16F10 melanoma cells in pseudometastatic subcutaneous or lung tumor models [39]. Additionally, CNTs were demonstrated to be good CpG delivery vehicles in CX3CR1GFP mouse models. First, functionalized single-walled carbon nanotubes (CNT-CpG) were examined and confirmed to be nontoxic. Secondly, this functionalization increased the absorption of CpG in vitro as well as in intracranial gliomas. CNT-mediated administration of CpG also increased the production of proinflammatory cytokines by primary monocytes. Surprisingly, a single intracranial injection of low-dose CNT-CpG eliminated intracranial GL261 gliomas in half of tumor-bearing animals by activating NK and CD8 cells. Furthermore, the surviving mice were protected from the recurrence of intracranial tumors, indicating the activation of long-term anticancer immunity. These findings have immediate implications for future CpG immunotherapy studies [49].

In another study, acid-functionalized MWCNTs (ox-MWCNTs) were coupled with hyperthermia therapy to treat breast cancer. EMT6 tumor-bearing mice were treated with ox-MWCNTs and local hyperthermia at 43 °C, which resulted in full eradication of the tumor and a considerable improvement in the mice’s median survival. In addition, there was an increase in the infiltration and maturation of DCs in mice. Furthermore, a considerable increase in tumor-infiltrating CD8+ and CD4+ T cells, as well as macrophages and NKs, was found in tumors treated with ox-MWCNTs–hypothermia combination therapy [50].

Nevertheless, SWCNTs have been proven to be antigen carriers capable of transporting antigens into APCs and eliciting humoral immune responses against weak tumor antigens. In this case, Wilm’s tumor protein (WT1) ligands, an upregulated protein in many human leukemias and cancers, were covalently attached onto solubilized SWCNT scaffolds to form SWCNT–peptide constructs. These constructs were rapidly absorbed by professional APCs (dendritic and macrophages) in vitro. Additionally, immunization of BALB/c mice with SWCNT–peptide constructs paired with immunological adjuvant elicited specific IgG responses against the peptide, but not against the peptide alone or in combination with the adjuvant, showing that the SWCNTs were not immunogenic [51]. 

Proteins that interact with smaller nanoparticles tend to preserve their structure far better than those that interact with bigger ones because smaller nanoparticles have a higher surface curvature, which limits the area of contact with the proteins [52]. Therefore, any interaction between the CNMs with the proteins may alter their functionality. Along the same lines, another investigation was conducted using OT-1 mice (mice in which CD8+ T cells developed a transgenic TCR specific for the SIIN peptide of ovalbumin displayed on H-2Kb). To circumvent the denaturing effects of their direct adsorption on CNTs, a simple yet robust technique of noncovalently attaching the T cell stimulus to the CNT substrates was developed. This demonstrated that CNT-based substrates can be designed to deliver MHC-I effectively for antigen-specific activation of T cells. They investigated the interaction of MHC-I with CNTs in a wide variety of other proteins to assess the stability and function of a physiological multimeric protein, MHC-I, on CNTs for applications linked to antigen-specific T cell activation. When compared with a soluble control under identical settings, the technique increased antigen-specific T cell responses by more than thrice. This study shed light on how noncovalent chemistry and adaptor proteins may be used to provide complex stimuli on CNT substrates [53]. 

When bundled SWCNTs are chemically treated to generate functionalized bundled SWNTs (f-bSWNTs), it improves protein adsorption compared with conventionally bundled SWCNTs. Indeed, f-bSWNTs have been discovered to be efficient antigen-presenting substrates. Splenocytes obtained from the spleens of C57BL/6 mice were treated with T cell antigens and costimulatory ligands (CD3 and CD28) adsorbed on these substrates to examine the kinetics of T cells’ responses on the surface of the nanotubes. The stimulation of primary T lymphocytes isolated from mouse spleens by these antibody-adsorbed substrates was measured by the cytokine secretion of traditional activation determinants such as interleukin-2 and interferon gamma (IFN-γ). The adsorption of T-cell-stimulating antibodies has been demonstrated to improve both the kinetics and amount of T cell activation. When compared with comparable artificial substrates with a large surface area and similar chemistry, this improvement is unique to f-bSWCNTs. These results supported the utilization of chemically processed nanotube bundles as an effective substrate for antigen presentation and indicate their potential utility in clinical applications requiring the presentation of artificial antigen [54]. 

Previously, Fadel et al. investigated the utilization of SWCNT bundles in the presentation of T-cell-activating antibodies to evoke immune responses against specific targets such as tumors. Because of the vast surface area of these bundles, T-cell-stimulating antibodies, such as anti-CD3, can be delivered at high local concentrations, resulting in powerful activation of T cells. Therefore, antibody stimuli adsorbed onto SWCNT bundles constitute a unique model for the effective activation of lymphocytes, with implications for fundamental science and clinical immunotherapy [55].

### 5.2. Graphene 

Yue et al. synthesized graphene oxide (GO) with an OVA (ovalbumin) antigen construct and tested the efficiency of its immune activation in E.G7-OVA tumor-bearing mice utilizing bone marrow dendritic cells (BMDC, primary professional cells for antigen presentation). In addition, the levels of the costimulator CD86 and the MHC II molecule were examined after GO-OVA uptake in vitro and found to be elevated. In an E.G7 tumor-bearing mouse model, tumor development was considerably inhibited in the GO-OVA group. Because of the two-dimensional graphene oxide’s unique bio- or physiochemical characteristics, GO-OVA increased cell recruitment, antigen transport, and antigen cross-presentation to CD8 cytotoxic T cells (GO). It also caused autophagy to be activated, which contributed to the programmatic activation of particular CD8 T cells in vivo [56]. 

GO functionalized by polyethylene glycol (PEG) and polyethylenimine (PEI) has been reported as a vaccination adjuvant for immunotherapy, with urease B (Ure B) as a model antigen, which is the particular antigen for *Helicobacter pylori* and has been recognized as a Class I carcinogen for gastric cancer. The treatment of DCs with GO-PEG-PEI significantly increased the production of interleukin 12 (IL-12), which is critical in the stimulation of NKs and T lymphocytes. Importantly, it accelerated the maturation of DCs and increased their cytokine release by activating several toll-like receptor (TLR) pathways. Furthermore, this GO-PEG-PEI worked as an antigen carrier, efficiently transporting antigens into DCs, implying prospects for cancer immunotherapy [57]. 

Wang et al. proposed a unique alum-based adjuvant formulation by creating AlO(OH)-modified graphene oxide (GO) nanosheets (GO-AlO(OH)), which, in addition to preserving the induction of the humoral immune response by AlO(OH), may also elicit a cellular immunological response by GO. A GO-AlO(OH) vaccine formulation was created by including the antigen using a simple mixing/adsorption method. In mouse models, antigen-loaded GO-AlO(OH) nanocomplexes increased antigen absorption and boosted the activation of DCs, eliciting greater antigen-specific IgG titers, producing a strong CD4+ and CD8+ T lymphocyte response, and suppressing the development of melanoma tumor [58].

### 5.3. Fullerenes

There is growing evidence that fullerene-based nanomaterials such as C_60_(OH)_20_ nanoparticles have antitumor activity [59]. Water-soluble C_60_(OH)_20_ nanoparticles have demonstrated effective antitumor immunomodulatory effects on immune cells such T cells and macrophages both in vivo and in vitro. For instance, they boosted the production of Th1 cytokines (IL-2, IFN-γ, and TNF-α), which helped to kill tumor cells through increased production of CD4+/CD8+ lymphocytes [60]. Another fullerene-based nanomaterial derivative is fullerol (C_60_(OH)_x_). C_60_(OH)_x_ was investigated for tumor-inhibitory activity in the H22 hepatocarcinoma mouse model. C_60_(OH)_x_ improved the phagocytotic activity of peritoneal macrophages. Additionally, C_60_(OH)_x_-treated macrophages generated more tumor necrosis factor alpha in vitro, implying that C_60_(OH)_x_ can boost innate immunity in tumor-bearing mice, thereby limiting the development of tumors [61].

### 5.4. CDs 

To achieve the desired cancer immunotherapy, polymer-coated CDs were evenly inserted into an ordered framework of mesoporous silica nanoparticles (CD@MSNs). The acquired CD@MSN was not biodegradable, but it could perform photothermal imaging-guided PTT in vivo. Interestingly, it was discovered that CD@MSN-mediated PTT could achieve immune-mediated prevention of tumor metastasis by promoting the proliferation and activation of NKs and macrophages while upregulating the production of cytokines such as IFN-γ and granzyme B. This study offered an unconventional method of producing biodegradable mesoporous silica and gave novel insights into the anticancer immunity associated with biodegradable nanoparticles [62].

To create chiral nanovaccines for cancer immunotherapy, researchers have used chiral CDs as carriers and adjuvants, as well as ovalbumin (OVA) as an antigen model. This design was efficiently internalized by bone marrow-derived DCs (BMDCs) from mice. Because of their fluorescence, chiral CDs could measure cellular absorption noninvasively and elicit a robust immunological response with increased BMDC maturation, T cell proliferation, and cytokine release. Second, it elicited a robust antitumor T-cell-mediated immune response and suppressed the development of B16-OVA melanoma tumors implanted in C57BL/6 mice. In vitro tests revealed that chiral CDs had a comparable capacity to LPS for inducing the maturation of BMDCs. This study proposed a novel method for producing multifunctional nanovaccines for enhanced cancer treatment [63].

As vaccine adjuvants, photoluminescent CDs were coupled with the model tumor protein antigen ovalbumin (OVA). These CDs greatly enhanced antigen absorption and the maturation of DCs. The CD–OVA nanocomposite dramatically enhanced the levels of the costimulatory molecules CD80 and CD86, which were uses as markers of DCs’ maturation. In addition, DCs produced more tumor necrosis factor (TNF-α). Furthermore, CD–OVA was demonstrated to significantly boost the proliferation of splenocyte and the production of IFN-γ. Interestingly, this CD–OVA vaccination was successfully endocytosed and processed by immune cells in vivo, resulting in significant antigen-specific cellular immune responses that inhibited the development of B16-OVA melanoma cancer in C57BL/6 mice [64].

### 5.5. Nanodiamonds

Another type of nontoxic CNM, fluorescent nanodiamonds (FNDs), was used to excite NKs and monocytes as an approach to boost antitumor activity. The absorption of FNDs and immune cell activation were significantly dose-dependent, as evaluated by the increased production of monocyte-derived TNF-α and NK cell-derived IFN-γ. Following subcutaneous injection, FNDs were seen in wild-type BALB/c mice [65].

To boost the DC-driven anti-GBM immune response, doxorubicin–polyglycerol–nanodiamond composites (nano-DOX), a potent inducer of DAMPs was created. In vitro, nano-DOX stimulated both human and animal DCs to inhibit glioblastoma cancer cells. Furthermore, nano-Dox promoted the infiltration and activation of mouse bone marrow-derived DCs as well as lymphocytes into glioblastoma xenografts. This suggested that administering nano-DOX through DCs might increase GC immunogenicity and elicit an anticancer immune response in GBM [66].

In summary, CNMs have emerged as a promising novel therapeutic platform to influence the immune response, particularly in cancer immunotherapy, due to their inherent characteristics and functionalization, targeted drug administration, and interactions with immune cells. CNMs may deliver carrier materials to particular cells, such as vaccines to APCs, to stimulate the immune system significantly. CNMs also permit stimulus responses in different cancer cells for combination cancer immune therapy. Furthermore, in order to activate the host immune response significantly and safely against cancer cells, CNMs have been engineered to transport antigens and chemotherapeutic agents to tumor cells. Through their combined application with chemotherapy, phototherapy, or radiotherapy, CNMs may elicit the activation of T cells against tumor cells and enhance anticancer efficacy with lesser toxicity [67]. Owing to the intrinsic high absorption of NIR by several CNMs, including CNTs and graphene-derived materials such as GO, the pairing of CNMs with phototherapy has gained popularity in recent years [68]. We have summarized the immunomodulatory properties of CNMs in preclinical cancer studies in Table 2.

## 6. Adverse Effects of CNMs: Lessons from In Vivo Models

The outcomes of research on the toxicological profiles of CNMs have shown both the cellular toxicities and immunological impact. We highlight below in Table 3 the reported adverse effects of CNMs based on in vivo tumor models.

Impurities, particularly catalyst metal contaminants such as Fe, Y, Ni, Mo, and Co added during synthesis and the purifying methods, contribute to CNTs’ toxicity. The existence of metal contaminants may result in contradictory findings about the biological properties, safety, and risk of CNTs, limiting their future practical uses [69]. For example, nickel oxide in SWCNTs may influence the redox characteristics of the regulatory peptide l-glutathione, a potent antioxidant that protects cells against oxidative stress [70]. Additionally, in one study, SWCNTs with varying metal contents were intratracheally administered into the lungs of spontaneously hypertensive rats (0.6 mg/rat) given once a day for two days in a row. This resulted in immediate and severe lung problems, including pulmonary inflammation, oxidative stress, and toxicity, as shown by the cell counts, MPO, LDH, albumin, protein, TNF-α, IL-6, MIP-2, CC16, and HO-1 data. Metal impurity-rich SWCNTs elicited much more negative reactions. After the injection of SWCNTs, the predominant lung histological abnormalities were pulmonary inflammation, multifocal granuloma development, and diffused CNT particle deposition in the alveoli, as well as bronchilocal cell hypotropy [71]. In another study, B6C3F1 mice were given 0.5 mg of raw or refined carbon nanotubes by intratracheal instillation. The CNTs caused dose-dependent epithelioid granulomas and, in some cases, interstitial inflammation, causing lung lesions. Some mice’s lungs also displayed peribronchial inflammation and necrosis that spread into the alveolar septa. In addition, fatigue, inactivity, and weight loss were reported 4 to 7 days after the CNTs were implanted. Because unprocessed nanotubes are so light, they might become airborne and potentially enter the lungs. CNTs may be far more dangerous after they enter the lungs, and are considered a serious occupational health issue in chronic inhalation exposures [72].

Using a different route of exposure, C57BL/6 mice were intraperitoneally injected with each CNTs, resulting in a total exposure of 50 g per mouse. A comparison of the inflammatory reactions of these numerous forms of CNTs, including three distinct types of MWCNTs and one type of SWCNT, revealed that peritoneal CNT injections of long and thick MWCNTs generated significant inflammatory effects. Furthermore, a sensitive approach for detecting DNA damage at the level of the individual eukaryotic cell was applied, which revealed considerable DNA damage in vitro [73]. It is also important to note that various CNMs, including SWCNTs, MWCNTs, and fullerene (C_60_), were evaluated for their toxicity. At a low dosage of 0.38 µg/cm^2^, SWCNTs dramatically inhibited the phagocytosis of alveolar macrophages (AM) derived from adult pathogen-free healthy guinea pigs, but MWCNT10 and C_60_ produced damage only at a high dose of 3.06 µg/cm^2^. Furthermore, macrophages treated with SWCNTs or MWCNT10 at 3.06 µg/cm^2^ displayed characteristics of necrosis and degeneration. This showed that the dose-dependent cytotoxicity mechanisms of SWCNTs and MWCNT10 are distinct [74]. Of note, the acute toxicity of C_60_ fullerene in mice was studied 14 days after a single intraperitoneal dosage was later on evaluated. C_60_ fullerene had no harmful impact in the dosage range of 75–150 mg/kg; the toxic effect of C_60_ fullerene was detected at concentrations of 300 mg/kg and above, and it was associated with behavioral disruption, hematotoxicity, and pathomorphological abnormalities in the spleen, hepatic, and renal tissues in mice. A dosage range of 75–150 mg/kg of a C_60_ fullerene aqueous colloid solution was found to be safe and might be used for biological reasons [75].

The immunological characteristics of oxidized water-dispersible MWCNTs in normal BALB/c mice were also examined after a subcutaneous injection of MWCNTs. The dynamic fluctuation in C3 and C5a levels in the serum suggested that this mode of delivery promoted activation of the complement quickly after the MWCNT injection. The MWCNTs activated the complement and produced proinflammatory cytokines such as interleukin (IL)-17, I-TAC, IL-1β, and IFN- γ early on. However, the complement and cytokines levels reverted to the baseline over time. There was no evident buildup of MWCNTs in the liver, spleen, kidney, or heart, with the exception of the lymph nodes. Histological examinations revealed just a modest inflammatory reaction at the injection site, with no granulomas identified over time. These findings contradicted previous findings when carbon nanotubes were administered intratracheally or intraperitoneally. Hence, these findings showed that administering MWCNTs subcutaneously was safer than administering them systemically [76].

Finally, the effects of GO and reduced graphene oxide (rGO) on glioma tumor cells directly implanted in models of chicken embryo chorioallantoic membranes were studied. The malignancies were removed after three days for additional examination. At a concentration of 100 µg/mL, increased quantities of GO and rGO resulted in reduced cell proliferation, viability, and cell organelle damage in glioma tumor cells. The findings showed that the interaction of GO and rGO with the glioma cells in tumors, which resulted in severe toxicity, was dependent on the shape of the graphene’s surface [77].

**Table 3 ijms-24-06600-t003:** Adverse in vivo effects of CNMs.

CNMs	Dosage	Exposure Method	In Vivo/In Vitro Model	Effect	Reference
SWCNTs	0.6 mg/rat	Nonsurgical intratracheal instillation	Spontaneously hypertensive (SH) rat	Pulmonary inflammation, multifocal granuloma formation, and a diffuse pattern of CNT particulate deposition in the alveoli as well as bronchilocal cell hypotropy	[71]
MWCNTs	50 µg per mouse	Intraperitoneal injection	C57BL/6 mice	DNA damage and severe inflammatory effects	[73]
MWCNTs	1.0 mg per mouse	Subcutaneous injection	BALB/c mice	Evident buildup of MWCNTs in lymph nodes	[76]
SWCNTs	0.38 μg/cm^2^	Cell culture	Alveolar macrophages(AM) from adult healthy pathogen-free guinea pigs	SWNTs greatly impaired the phagocytosis of AM at the low dose of 0.38 μg/cm^2^	[74]
CNTs	0.5 mg per mouse	Intratracheal instillation	B6C3F1 mice	Epithelioid granulomas and, in some cases, interstitial and peribronchial inflammation, and necrosis	[72]
C_60_ fullerene	300 mg/kg of C_60_ fullerene	Intraperitoneal injection	Mice	Behavioral disturbances, hematotoxicity and pathomorphological changes in spleen, hepatic, and kidney tissues	[75]
GO	100 μg/mL GO solution	Direct injection	Chorioallantoic membrane of fertilized chicken embryos	Decreased vitality, reduced cell proliferation, damaged cell organelles	[77]

## 7. Translational Challenges of CNMs in Oncology

Given the promising data of CNMs, alone or as drug carriers, to modulate the immune response, as discussed above, they have great potential for clinical applications, such as cancer therapy. However, there are important concerns and challenges. In Table 4, we summarize the current issues to be overcome before their successful translation from the laboratory to the clinic.

Primarily, because the long-term consequences of CNMs are time-consuming to investigate, they have rarely been documented. Next, additional research into a more precisely regulated production procedure for some of these CNMs is still required. Materials derived by various synthesis techniques typically have highly varied characteristics and, as a result, distinct biomedical properties. For example, graphene with a single or few layers is typically required for biological applications that require a more regulated production technique [78].

A variety of surface modifiers and biomacromolecules have been created to enhance the characteristics of CNMs. The functionalization improves the efficacy of their application in the field of biomedicine such as immuno-oncology [79]. However, some functionalizing agents may produce undesired effects in the process. For example, PEG is commonly used to functionalize CNMs such as GO. PEG has been shown to be immunogenic, interfering with the effects of the given antigen [80]. Another aspect to consider is the presence of pre-existing anti-PEG antibodies in the blood of some healthy donors. In fact, anti-PEG antibodies have been proven to affect the therapeutic effectiveness and safety of PEGylated medicines [81]. 

Additionally, the size, shape, and chemical surface composition of several nanomaterials can determine the impact of their immunological regulation. This creates the unprecedented unpredictability of any new potential nanomedicine in any clinical trial [82]. This implies that there could be inconsistent results in clinical trials in cases where the size, shape, or surface compositions are slightly distorted. 

Nanomaterials can be used not only as drug carriers but also as immunomodulators of certain biochemical processes. This poses a difficulty in measuring the effects induced by either the CNM or the antigen being delivered. Furthermore, polyhydroxylated fullerenols were reported to have immunosuppressive effects on macrophages and T cells. Such effects included tilting the cytokine balance, favoring the release of Th1 cytokines and reducing the secretion of Th2 cytokines [83]. 

In a study conducted by Schrand et al., CNMs demonstrated both material- and cell-specific cytotoxicity. In fact, a general trend for biocompatibility with the susceptibility of macrophages to cytotoxicity involving nanodiamonds, MWCNTs, SWCNTs, and carbon black particles was found. Indeed, macrophages were shown to be more susceptible to cytotoxicity compared with neuroblastoma cells [84]. Because malignancies develop in a variety of cells across the body, this might be a significant translational hurdle for clinical trials.

**Table 4 ijms-24-06600-t004:** Benefits and challenges of CNMs.

CNM	Benefits	Challenges	Reference
CNTs	Tunable physical qualities, biocompatibility, and a large surface area are available. Mechanical properties, aspect ratio, conductivity, and chemical stability are all excellent	Lack of aqueous solubility, nonhomogeneous size (diameter and length), and the possibility of metallic contaminants. Some CNTs are lightweight powders, which may penetrate the respiratory tract	[85]
Graphene	It has outstanding electrical, optical, and thermal characteristics. Graphene’s two-dimensional atomic sheet structure allows for more diversified electronic properties than CNTs	Colloidal instability, lack of consistency, inadequate synthetic control, low chemical stability in biological environments, oxidative susceptibility	[86]
Nano diamonds	Fluorescence and photoluminescence, biocompatibility, and lower size when compared with other CNMs; hardness; corrosion resistance; chemical inertness; strong electrical resistance; and optical transparency	Difficult to make covalently, time-consuming to remove harmful chemical solvents after production, sudden drug release, and proclivity to aggregate	[87]
CDs	Excellent photo- and electro-catalytic characteristics, unique photoluminescent properties	High toxicity, low biocompatibility, high cost, and low chemical inertness	[88]
Fullerenes	Good photoelectrochemical qualities, the possibility of surface modification, and superconductivity	Low aqueous solubility, accumulation in the cell membrane, susceptibility to breaking down in the presence of light and oxygen, susceptibility to deactivation processes	[89]

## 8. Future Perspectives

Thanks to their unique physicochemical properties and, perhaps more importantly, their interactions with the immune system, CNMs have offered new approaches for the enhancement of immune-based therapies against cancer. In this review, we have discussed the immunomodulatory mechanisms of CNMs and have highlighted the current status of preclinical applications of CNMs in cancer therapy (Table 2). Before the full potential can be achieved for this therapeutic modality, we have highlighted the adverse in vivo effects, as well as translational challenges of CNMs in oncological studies (Table 3 and Table 4). One of the attractive properties of CNMs is their use as effective platforms for drug delivery and targeting. The flexibility of chemically modifying CNMs provides another layer of the capacity of CNMs to create new systems that can be adapted to specific interventions. For example, CNMs have been used for targeted drug delivery to enhance the efficacy of other treatments, such as chemotherapy. In addition, the covalent and noncovalent functionalization of CNMs with different biomolecules, drugs, or antibodies allows their selective accumulation in tumors. Therefore, one of the directions in the future is the combinations of CNMs and current treatment regimens, although this is still in the early stage. To move forward, one challenge is to identify the population of patients who are most likely respond to the therapy. In this regard, further investigation is required to identify the biomarkers that predict the maximal synergy of CNMs with the current therapies. In the near future, it will be interesting to see the discovery of novel biomarkers for the therapeutic response of CNM-based combination treatments and the best approaches to manipulating the immune response in favor of antitumor benefits as medicine moves towards the development of custom-tailored precision therapies.

## Figures and Tables

**Figure 1 ijms-24-06600-f001:**
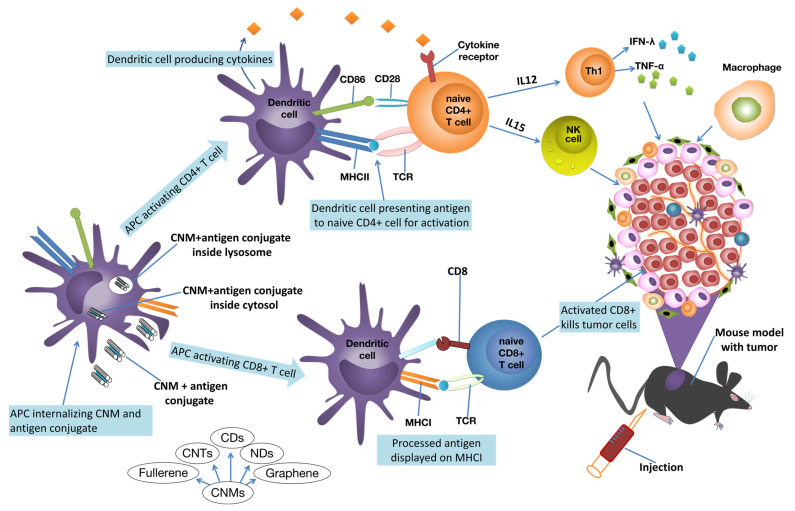
Schematic representation of the immunomodulatory effects of CNMs in cancer therapy. CNMs enhance antitumor immunity through multiple and diverse mechanisms of immune modulation. Antigen-presenting cells (APCs), namely dendritic cells (DCs), pick up the CNM–antigen conjugate and transfer the antigen peptides to naive T cells for activation. A multitude of unique receptors on the surface of DCs serve as natural recognition sites for activating certain immune cells. Nevertheless, targeting DCs alone is insufficient to elicit a significant immunological response. The movement of antigens to particular compartments for presentation in DCs is critical. In DCs, for example, the lysosome-dependent route leads to the antigen breaking down into antigenic peptides within the lysosomes, which are then loaded onto Class II major histocompatibility complex (MHC-II) molecules for presentation to CD4+ helper T cells. MHC-I molecules, on the other hand, display cytosolic antigens to activate CD8+ T cells and trigger cytotoxic T lymphocyte (CTL) responses [45]. Other cytokines, such as TNF- α, send chemical signals to the tumor, causing inflammation and cell death [46]. IFN-γ is largely released by activated T cells and natural killer (NK) cells, and has the ability to activate macrophages and improve antigen presentation [47]. Many cytokines, notably IL-15 and IL-12, can activate and stimulate proliferation and expansion the of NK cells, as well as other antitumor immune cells, including CD8+ T cells [48].

**Table 1 ijms-24-06600-t001:** Classification and main characteristics of CNMs.

Dimension	CNMs	Main Characteristics	References
0D	Fullerenes Particulate diamond Carbon dots (CDs)	Materials on the nanoscale in all dimensions	[9,14]
1D	Carbon nanotubes (CNTs: SWCNTs, DWCNTs, MWCNTs) Carbon nanofibers (CNFs) Diamond nanorods	Materials have one dimension that is larger than nanoscale	[9,14]
2D	Graphene Graphite sheets Diamond nanoplates	Thin-sheet materials of nanoscale thickness	[9,14]
3D	Nanostructured diamond-like carbon (DLC) films Nanocrystalline diamond (NCD) films Fullerite	Multilayer materials composed of several building pieces, including 0D, 1D, and 2D CNMs	[9,14]

0D, zero-dimensional; 1D, one-dimensional; 2D, two-dimensional; 3D, three-dimensional; CDs, carbon dots; CNTs, carbon nanotubes; SWCNTs, single-walled carbon nanotubes; DWCNTs, double-walled carbon nanotubes; MWCNTs, multiwalled carbon nanotubes; CNFs, carbon nanofibers; DLC, diamond-like carbon; NCD, nanocrystalline diamond.

**Table 2 ijms-24-06600-t002:** Immunomodulation of CNMs in preclinical cancer studies.

CNM	Immune Cells	Molecules Delivered	Cancer Type	Results	Reference
MWCNT	CD4+/CD8+ T cells, bone marrow−derived DCs	Anti-CD40 Ig, CpG, and OVA	B16F10 melanoma cells	In vitro: enhanced T cell responses to OVA antigens, increased IFN-γ production In vivo: reduced development of tumor cells expressing OVA	[39]
SWCNT	DCs, macrophages, CD4 T cells	Wilm’s tumor protein, WT1 peptide 427	Human leukemia and tumors	In vitro: increased the rate of antigen absorption by APCs (DCs and macrophages) In vivo: SWCNT–peptide conjugates stimulated peptide-specific IgG immunological responses and IFN secretion in CD4 T cells;	[51]
SWCNT	Bone marrow-derived monocytes from mice, NK cells, CD8+ T cells	Oligonucleotides (CpG)	Intracranial GL261 gliomas	In vitro: increased CpG absorption, increased proinflammatory cytokine secretion in primary monocytes In vivo: suppressed intracranial GL261 gliomas by activating NK and CD8 cells	[49]
SWCNT	Primary splenocytes, primary T cells	CD3, CD28, T-cell antigens	Multiple cancers	In vitro: increased antigen-presenting capacity, enhanced cytokine secretion in T cells	[54]
SWCNT	T cells	anti-CD3	Multiple cancers	In vitro: increased T cell activation	[55]
GO	CD4+/CD8+ T cells, bone marrow derived DCs	Ovalbumin (OVA)	B16 murine melanoma tumor cells	In vitro: increased antigen absorption and stimulated CD4+/CD8+ lymphocyte responses In vivo: suppressed tumor growth	[58]
Fullerenes	Macrophages, CD4+/CD8+ T cells, NK	C_60_(OH)_20_ nanoparticles	Abnormal cells	In vitro: enhanced production of antitumor cytokines (TNF-α, IFN-γ, IL-2) In vivo: suppressed tumor growth	[60]
Fullerene	Macrophages	C_60_(OH)*_x_*	Murine H22 hepatocarcinoma	In vitro: activated peritoneal macrophages, increased TNF-α production In vivo: suppressed tumor development	[61]
GO	DCs, splenocytes	Urease B GO-PEG-PEI	Gastric cancer	In vitro: increased the production of interleukin 12 (IL-12); In vivo: increased T cell proliferation	[57]
CD	DCs, splenocytes	Ovalbumin (OVA)	Melanoma	In vitro: increased TNF-α levels in DCs, activated T cells to produce more IFN-γ, increased DC maturation In vivo: suppressed tumor development	[64]
ND	DCs, U87 MG cells	Doxorubicin	Glioblastoma	In vitro: activated DCs In vivo: enhanced the infiltration and activation of DCs and lymphocytes	[66]
ND	Human NK cells and monocytes, murine macrophage cells,	FNDs	Multiple cancers	In vitro: uptake of fluorescent ND by RAW264.7, NK, NKL, and monocytes In vivo: fluorescent ND observed in mice	[65]
GO	Bone marrow-derived DCs, CD8+ T cells	Ovalbumin (OVA)	E.G7-OVA tumor (mouse lymphoma cells)	In vitro: increased OVA uptake attached to GO by BMDC, increased expression of CD86, MHC I, and MHC II; In vivo: reduced tumor volume	[56]

## Data Availability

Not applicable.

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
