# Peer review of "Carbon Nanomaterials: Emerging Roles in Immuno-Oncology"

_ijms, 2023, doi:10.3390/ijms24076600_

Round 1
Reviewer 1 Report
The study is difficult to read. Consists of a compilation of many studies that report some efficacy in combination with other immunotherapies.
The study should be organized different.
1. Discuss how carbon nanomaterials can reach solid tumors ? Have they been used for targeting hematologic tumors?
2. Discuss safety, tolerability and efficacy in in vivo tumor models, especially to repeated dosing of carbon nanomaterials. Are their unwanted immunogenic effects to repeated dosing? Adverse side effects as lung inflammation and fibrosis or ROS production and DNA damage?
3. Discuss if carbon nanomaterials based on literature have a real chance for clinical application in the future as a monotherapy or in combination in patients with cancer.
Give only few examples of literature. Not copy the results of many manuscripts into one paragraph without real structure and discussion.
Reviewer 2 Report
In this manuscript, the authors reviewed the anti-tumor applications of carbon nanomaterials in preclinical and clinical studies. They also describe the potential anti-tumor immune mechanisms of these materials. Overall, this manuscript should be beneficial to readers working on biomedical applications of carbon nanomaterials. Detail comments and suggestions regarding the manuscript are described below.
1. It is suggested that the authors indicate the carbon nanomaterials used in the study mentioned in each paragraph in Section 4 for clarity.
2. Section 4.1.1: The anti-tumor immune mechanism described here is not only relevant to CD4+ T cells, but also associated with CD8+ T cells. It is suggested that the authors combine section 4.1.1 and 4.1.2 together.
3. Section 4.1.2, 4.1.3, and 4.2.2: Some of the materials described in these sections are not relevant to carbon nanomaterials, the authors may need consider to remove these references in the manuscript.
4. Section 4.2.3: The anti-tumor immune mechanisms described in the examples provided in this section is irrelevant to natural killer cells. It is suggested that the authors remove this section.
5. Page 11, third paragraph: The meaning of “THP-1 macrophage differentiated cells against human neutrophils and HL-60 neutrophil differentiated cells” is unclear. Additionally, “very small number of neutrophils were up taken by MWCNTs” is not reasonable.
6. The number of references is not in the correct order in the manuscript. Some of the references, such as Ref 10-17, were missing. Additionally, the numbers of the references were wrong after the Section 4.
7. Some of the space between two words were missing in Table 2
8. It is suggested that the authors did the English editing before submitting the revised manuscript.
Reviewer 3 Report
Dear authors,
The subject of your review is interesting but I have a few recommandations to try to improve the structure and maybe challenge a bit further your work:
1/ In the discussion (you only have perspectives, I don't know if it is a criteria from the journal but I believe a discussion in which you can add the persepectives is missing) I believe the reader would like for you to compare all these roles in immuno-oncology to the ones of other nanoparticles such as lipidic. Is there any work comparing the two, does the inorganic composition stimulate the IS more ? Considering the challenges you expose why working on these and not polymeric or lipidic nanoparticles that can already be found in the market ?
2/ In table 1 make sure that all the nanoparticles you describe further are in it (e.g., MWCNT isn't specify, although I am guessing you include it in CNT).
3/ Update in preclinical and clinical cancer studies. It is unusual to describe clinical studies before pre-clinical. Moreover most of the exemples presented in Table 2 do not apply in oncology, so I am not sure it should be highlighted that much. I believe maybe you should first discuss preclinical studies (Table 3 with the different mechanism then the applications in oncology (which there is just a few) and explain that this is because of the translationnal challenges you expose further on.
4/ Considering all the other applications you mentionned, I would also discuss what makes oncology a complicated field for these carbon nanoparticle ? Is it the desease, the repeated administration, the injection route ...
5/ Table 2: SInce most of the examples are not in oncology I wouldn't keep this table but if you choose to keep it maybe you should change the column "Title" by 'application' and describe inyour words this application. COuld be more understandable.
6/ Table 3: Is the heart of your work. To make it easier for the reader I would add a column "Model" where you specify in vitro/ in vivo (animal) instead of writing it in the results (for some it isn't specified).
7/ I believe Table 3 and all the presented immunoregulatory mechanisms should be linked and therefore be together in a same section. And I don't understand why some references suche as 49, 50, 56 and so on ... aren't in Table 3 ?
8/ This mechanism part is very dense and long to read, thus I would recommand to make a figure more complex than Figure 1, that show the immune cells, immune effectors, CNMs and mechanism of action.
Round 2
Reviewer 2 Report
The authors have responded to the questions raised by this reviewer.